# Challenges of Caregivers Regarding Homecare to Type 1 Diabetic Children in Vhembe District, South Africa: A Qualitative Study Report

Margaren Ndou, Ndidzulafhi Selina Raliphaswa * and Azwidihwi Rose Tshililo

Department of Advanced Nursing, Faculty of Health Sciences, University of Venda, Private Bag X5050, Thohoyandou 0950, South Africa
* Correspondence: ndidzulafhi.raliphaswa@univen.ac.za

**Abstract: Background:** Worldwide, type 1 diabetes mellitus disease is a devastating health condition for both the diagnosed children and caregivers taking care of them. These challenges were drastically affecting caregivers in rendering the required homecare service for their patients. Caring for a child with diabetes mellitus often hurts the caregivers and other family members. **Aim:** This study aimed to explore the challenges experienced by caregivers during the provision of care to type 1 diabetic children. **Setting**: The study was conducted in selected health facility of Vhembe District, South Africa. **Methods**: A qualitative design that was both descriptive and exploratory was used. Non-probability purposive sampling was used to select the caregivers who were taking care of children with type 1 diabetes mellitus at their homes. Semi-structured interviews were conducted with 15 participants from the Vhembe district of Limpopo province. **Results:** Caregivers experienced various challenges related to re-admission, poor understanding of medication, low level of literacy, committed mistakes, poor adherence, and fear of giving an injection. **Conclusion**: It is imperative to educate the caregivers on the care of children with type 1 diabetes mellitus for them to be competent and knowledgeable in assisting their diabetic children at home.

**Keywords:** caregivers; challenges; children; experiences; type 1 diabetes mellitus



## 1. Introduction

Caregivers have the responsibility of providing homecare to children with type 1 diabetes mellitus (T1DM). They are required to provide treatment and general psychological support during their developmental stages. T1DM is a chronic disease associated with abnormally higher levels of sugar (glucose) in the blood [1]. T1DM is classified by the higher or lower abnormal insulin levels produced by the pancreas, which releases insulin to help in the transportation of food to body cells. The World Health Organization [2] indicated that about one million children in the world will have been affected by T1DM-disease by 2020 and 366 million by 2030.

Caregivers experience challenges in relation to the treatment and management of children with T1DM, as they are not professionally trained. The study by [3] revealed that caregivers in most developing countries experience challenges when dealing with children with T1DM; these challenges include: the routine monitoring of blood glucose levels, preparing and maintaining a balanced nutrition plan, and overseeing the children's physical activities.

According to [4], caregivers of children with T1DM experience emotional distress at the responsibility of taking care of a child with T1DM at home. These include thinking of the reluctant and naughty children who could be forced by their peers to eat forbidden fruits immediately after being warned. Unfortunately, this behavior usually affects their health and education negatively. Increased glucose level results in ketoacidosis (DKA), leading to admission, which has consequences on the child's progression at school. However,

ref. [5] emphasizes the importance of education and behavioral support, which could benefits caregivers by equipping them with skills to address their challenges when caring for their children with T1DM. The other challenges they encounter daily might include a daily routine of administering medication, diet, daily physical activities, and lifestyle modification. In support of this view, ref. [6] also noted that the required lifestyle changes also bring about extended stress levels to the caregivers, because of re-admission, the decline in health, gradual immune system functionality, and the decline in weight that affect children diagnosed with T1DM. The study done by [7] is in support of the latter as it highlighted that inadequate sleep has a negative impact in children because it predisposes them to obesity due to increased weight gain. It is in this light that the study explored the challenges experienced by caregivers during the provision of care to type 1 diabetic children.

## 2. Materials and Methods

### 2.1. Study Design

A qualitative design that was both descriptive and exploratory in nature guided the study in achieving the objectives of the study through understanding how caregivers in the Vhembe district of the Limpopo province experience challenges when caring for children with T1DM. The Consolidated Criteria for Reporting Qualitative studies (COREQ) (in Supplementary Materials) [8] was used to craft this sub-section below, demonstrating how this paper addresses issues of rigor, representation, and, reflexivity.

Since little was known about the experiences of these caregivers, an exploratory design was adopted for this very reason [9]. Similarly, the descriptive nature accurately described the experiences of caregivers regarding the home care of children with T1DM as narrated by caregivers. The study adopted non-probability purposive sampling to select the caregivers who were taking care of children with T1DM at their homes. Only those whose ages ranged from 18 to 57 years and were willing to participate were included. However, those who were not mentally stable were excluded even if they met the criteria. A sample of 15 participants was chosen as a suitable sample with data saturation occurring at 10 participants. However, the researcher went on with the interviews to ensure that nothing new was coming in.

### 2.2. Setting

The study was conducted in deep rural areas of Vhembe District, which lies in the northern part of Limpopo Province, which is one of the nine provinces in South Africa. Vhembe covers 18,569 square kilometers and a population of 1.3 million people. The cultural population that resides in Vhembe District are the Vhavenda and Tsonga people. Vhembe is one of five districts in Limpopo Province and is comprised of four local municipalities: Makhado, Thulamela, Musina, and Collins Chabane. There are six district hospitals in Vhembe. These are Siloam Hospital, Memorial Hospital, Elim Hospital, Malamulele Hospital, Donald Frazer Hospital. and Musina Hospital. In Vhembe district, there is only one referral hospital, called Tshilidzini. This hospital serves communities around Siloam, Elim, Malamulele, Donald Frazer, Memorial, and around Tshilidzini and Musina. This study was conducted at homes of children with type 1 diabetes who were re-admitted at Tshilidzini Referral Hospital.

### 2.3. Data Collection

An unstructured in-depth individual interview was administered with the aim of obtaining full information from the participants who had first-hand experience regarding the topic under study. The first female author, who is a master student in nursing, conducted an interview. The author is a registered pediatric nurse with more than 10 years' experience working with children. All information related to the study was discussed with the participants. This was done in order to let participants know exactly the kind of research to be done and the rationale for doing the research. This was also a way of building a rapport with the participants and for them to gain a full understanding. Researcher's interests and personal goals regarding the topic were highlighted. Pretest was done before the actual

study. This was done to identify any flaws related to the questioning, probing skill, and clarity of the question. Those who were part of the pretest were not included in the main study. The opening central question was asked: '*What are your experiences regarding home care of children with T1DM?*' Subsequently, participants were encouraged to talk during an interview through probing techniques that involved making vague comments that could have multiple meanings to obtain in-depth information. The researcher documented and audio-recorded each interview session after receiving consent from the participants. Data collection information included demographic, knowledge, and experiences of caregivers taking care of children with T1DM. Different sessions were conducted for each interview at various locations determined by the participant as a measure to create a conducive environment that allowed each participant to be comfortable and speak freely when there were no other people present. Field notes were taken during and after the interview to ensure that both verbal and non-verbal data were captured. Each interview lasted for 30 to 45 min. The audio-recorded data was transcribed verbatim, using the participants' language. The transcripts were sent back to the participants for correction and to confirm if what had been transcribed was a true reflection of what the participants said.

*2.4. Data Analysis*

The collected data were processed and analyzed using Tesch's open coding method of analysis in some orderly, coherent fashion so that patterns and relationships were discerned into quotes from filed notes, themes, and sub-themes directly from the transcripts. It is noteworthy that the transcription was coded by an independent contracted coder without the aid of software.

*2.5. Trustworthiness*

The trustworthiness of the study was ensured using various strategies, as suggested by [10]. Credibility was ensured by using prolonged engagement with the caregivers to fully understand the experiences they faced relating to the homecare of children with T1DM. The researcher spent at least an hour with the caregivers to build trust by explaining the purpose of the study and the ethical issues involved clearly. Essentially, while conducting the interviews at the caregivers' homes, the researcher used both English and a local language to communicate with the caregivers based on the home environment they lived in. These measures assisted in ensuring the triangulation of data, which helped in identifying the common characteristics and elements that were relevant to the caregivers' experiences. The researcher addressed the issue of dependability by ensuring that the reliability of the field notes and audio recordings were cross-referenced. The accuracy of the audio recordings and the field notes taken were cross-checked against each other over various time intervals.

The present study also ensured transferability by describing in detail the background and the home environment of the caregiver including verbatim quotations expressed during the interview. Conformability was ensured by replaying the audio recordings to the caregivers for them to verify that what had been recorded was a true reflection of their experiences. Ethical standards were adhered to. Approval by the Human and Clinical Trails Research Committee, University of Venda, and the provincial department of health (Project number: SHS/20/PDC/32/3107) was received. Similarly, permission for data collection was granted by the chief executive officer of the Tshilidzini hospital. Informed consent was obtained from the caregivers before any appointments could be scheduled. It is noteworthy that all approved ethical conduct and practices were observed throughout the entire study.

## 3. Results

*Demographic Profile*

The demographic information of the 15 participants provided a broader description and representation of the study as presented in Table 1.

**Table 1.** Demographic profile.

| A Total Number of Caregivers Interviewed: | Statistics of Participants per Demographic Data |
|---|---|
| **Age range** | |
| 18–24 years | 5 |
| 25–34 years | 4 |
| 35–44 years | 2 |
| 45 years and above | 4 |
| **Marital status** | |
| Single | 14 |
| Married | - |
| Widowed | 1 |
| **Level of education** | |
| Illiteracy | 2 |
| Below matric | 6 |
| Matric | 7 |
| **Source of income** | |
| Government grant | 13 |
| Local street vendors | 2 |

Using Tesch's open-coding method, Table 2 shows the themes were developed before carrying out of the data analysis and interpretation processes from the responses as given after the interviewing of the sampled participants (P).

**Table 2.** Themes and sub-themes of the experiences of home caregivers of children with T1DM.

| Themes | | Sub-Themes | |
|---|---|---|---|
| 1. | Lack of adherence to scheduled clinic appointments and medication instructions | 1.1 1.2 1.3 | Loss to follow-up Frequent admission of the T1DM child Failure to abide by medication instruction |
| 2. | Poor understanding of medication and health instructions | 2.1 | Confusion with the prescribed medical and healthcare practices |
| 3. | Lack of adherence to a diabetic diet, medication instructions and health education | 3.1 3.2 | Caregiver knowledge of a diabetic diet The attitude of the caregiver and T1DM child |
| 4. | Fear of administering insulin injections to children | 4.1 4.2 | Fear of injecting a child The fear of harming the child |

T1DM: Type 1 diabetes mellitus.

According to [11], caregivers looking after children with T1DM from a global view context appears to display a rise in mental disorders, impacting socio-economic status, health, and human rights. The results from the data analysis revealed four major themes that were discussed in detail below.

**Theme 1: Lack of adherence to scheduled clinic appointments and medication instructions**

Caring for a child with diabetes mellitus-disease often has a negative effect on the mental health of caregivers and other family members. Furthermore, this is the other issue that differentiates this pandemic disease from the other type of diseases, because besides the diabetic-sick children, it also affects those who are taking care of them (caregivers). This has resulted in varied individual perceptions about how one should apply the health, diet, and medication instructions to the provision of care for T1DM children.

**Sub-theme 1.1: Loss to follow-up**

Caregivers do not understand that they can manage and care for children with diabetes by simply following the prescribed caregiving responsibilities. Since many children rely on the medication that is provided by the hospital, it became a problem when they experience

complications at home due to caregivers not fulfilling their roles. This was confirmed by a caregiver who mentioned:

*"Yes, when looking after the T1DM patients, both the primary and secondary caregivers should work together (caregivers and professional health caregivers). We regularly attend check-up programs where professional caregivers teach us rules for diet and making exercises. I think we should respect each other's roles while looking after the T1DM patients. Ok! We should then work together as a solution towards securing the lives of these patients."* [P.14]

**Sub-theme 1.2: Frequency admission of T1DM**

It was realized that caregivers rely on clinics and hospitals to collect medication. However, failure to adhere to the prescribed medication may result in the readmission of some T1DM children. Similarly, other caregivers wait until the development of complications rather than adhering to their scheduled dates for routine check-ups. During the interviews, one participant said*:*

*"Ok! Yes, I have been looking after my daughter's sick child since birth, she has been quite adherent to the prescribed foods and fruits to eat but started to ignore them at the age of 10 years. By the way, this leads her to be always being readmitted at the hospital."* [P.7]

**Sub-theme1.3: Failure to abide by medication instruction**

It is still a challenge for many caregivers to provide the necessary healthcare practices and habits in caring for children with diabetes. These challenges often create a compromising situation for both the caregiver and the child when caring for and treating diabetes. The below statement is evidence of the need for the proper caretaking of children with T1DM:

*"Yes, Ok! I am going to defend myself, but the truth is that we both make mistakes. For example, a caregiver does sometimes give a patient the wrong medicine. On the other hand, the patient also eats the forbidden foods or fruits after being forced by the other peers. Hmmm! We all do whatever we can to help each other so that the patients stay healthier."* [P.8]

**Theme 2: Poor understanding of medication and health instructions**

Essentially, the health education provided by the health professionals to the caregivers is also of paramount importance because it helps to always keep remembering the rules to be followed while giving food or even injecting the T1DM children with some treatments.

**Sub-theme: Confusion with the prescribed medical and healthcare practices**

Caregivers of children with T1DM should always be encouraged to adhere to the medication prescribed by healthcare professionals. Caregivers should know the contraindications of their medication and should always report the side effects. One caregiver highlighted the following remarks:

*"I always attend the check-up programs where the professional healthcare workers teach us how to apply medications and follow the health rules. However, immediately thereafter, I would have forgotten everything but my relatives helped me many times."* [P.9]

**Theme 3: Lack of adherence to a diabetic diet, medication instructions, and health education**

Non-adherence to prescribed medication regimens is common in patients with caregivers of diabetic children due to the dietary and medication instructions being taken lightly by caregivers.

**Sub-theme 3.1: Caregiver knowledge of a diabetic diet**

The issue of diet had been stressed throughout the interview sessions. Knowledge of the suitable diet for children with diabetes was stressed. During the interviews, one caregiver highlighted the statement below:

*"Children are children, and nobody could predict their reactions well. A granddaughter, while still young used to obey the prescribed diet rules, but later ate the forbidden foods and fruits while with friends at school. Hmmm! The re-admission at the hospital was the result thereof."* [P.15]

### Sub-theme 3.2: Attitude of caregivers and T1DM children

The following caregiver reiterated the importance of adhering to medical instructions that prevent further complications and hospital readmission. In answering the posed question, the interviewed participant replied:

> *"A large group of the T1DM children does not usually adhere to the diabetic diet, medication instructions, and health education, which could lead them to some terrible complications while looking after the sick children at their homes. Some sick children even end up being readmitted at the hospitals due to increased glucose level which is not controlled".* [P.2]

### Theme 4: Fear of administering insulin injections to children

The physical and psychological demands of caring for children with diabetes may be impacted by the caregiver's challenges in their provision of care [12]. Administering different medical instructions and suggestions from professional health workers may require more than educational understanding from caregivers, as the adequate understanding and application of the medical and health instructions could play a vital role in improving the condition of the T1DM patients under care.

### Sub-theme 4.1: Fear of injecting a child

Some caregivers mentioned that it is not easy to apply the prescribed medication with the emphasis on the injection type of medication. The fear of hurting the child receiving the injection often creates reluctance in following the prescribed medication. Hence, one caregiver mentioned that:

> *"Yes, honestly speaking, my child is afraid of the injection but only wants to take tablets. Hmmm! It is indeed a problem, but there is presently nothing I could do". You know, I also feel uncomfortable injecting him because he cries a lot before and after injection. I am always guilty especially when he asks for forgiveness not to be injected. Eish, it is not easy . . . ."* [P.11]

### Sub-theme 4.2: The fear of harming the child

Caregivers do not understand that they can manage and care for children with diabetes by simply following the prescribed caregiving responsibilities. Since many children rely on the medication that is provided by the hospital, it becomes a problem when they experience complications at home due to caregivers not fulfilling their roles. This was confirmed by a caregiver who mentioned:

> *"Yes, when looking after the T1DM patients, both the primary and secondary caregivers should work together (caregivers and health care professionals). We regularly attend check-up programs where healthcare professionals teach us rules for diet and making exercise. I think we should respect each other's roles while looking after the T1DM patients. Ok! We should then work together as a solution towards securing the lives of those patients."* [P.14]

## 4. Discussion

Looking after T1DM patients is not an easy task but a very difficult one, which also needs more hands (more people) while doing it. Caregivers might do their caring duty properly if they could work together with other healthcare professionals [13]. On the other hand, caregivers who attend check-up sessions regularly, could be assisted in gaining adequate knowledge and understanding concerning caring for sick children [14]. From my own perception, it seems caregivers do not intend to do this in time, as they only waited until a T1DM illness problem surfaced. Yes, it is, of course, a reality that the secondary caregivers (health professional workers) are indeed helping them in terms of solving the T1DM patients' ailing conditions. Ideally, it should often be realized that problems will keep on happening to an extent that caregivers could start looking for other help from health professional workers. For example, a caregiver can keep fearing injecting her T1DM patient as explained by the health professional work during the check-up programs. In this instance, it means that the service of another co-worker is always needed. The

reason is that nothing could sound possible, unless a person who does not suffer from forgetfulness and is a professional health worker, can be used. Moreover, it also appears that if professional health workers had been used at such a pace, many unnecessary deaths of our young generation would have been avoided, bearing in mind that the secondary caregivers (doctors, nurses, and dieticians) are the best stakeholders to be easily used by the caregivers during the provision of care to the T1DM patients, since they appear to be always in a position of the required knowledge.

Time and again, T1DM children are being readmitted to the hospitals due to a failure to adhere to the scheduled foods and medication instructions. Furthermore, both caregivers and the so-called sick children (DM patients) should also be blamed. The reason is that both somehow or somewhere fail to abide by the prescribed instructions as taught by the doctors and dieticians. For example, in cases where the caregiver is an elderly grandmother, she might also forget to inject the sick child or even give the T1DM patient the wrong medication. On the same note, this may also lead to the very child becoming sick or sometimes being readmitted to the hospital. However, this serious action might occur after an elderly caregiver has failed to adhere to the rules taught to her by professional health workers. Consequently, that could lead the caregiver's grandmother to become more stressed. For that matter, if such a situation is not taken care of, it might drive a caregiver to be admitted to the hospital for no apparent reason. In short, both a sick child and the caregiver may, of course, be blamed for being the cause of some readmission cases. The only cause of such actions might be that both would have failed to adhere to the prescribed instructions from the professional health workers [15]. Therefore, it is important that the child, parents or family members, health care professionals, and even schoolteachers to work collaboratively, since disease management requires interdisciplinary care at an earlier stage [16].

Mistake after mistake is made not only by one person but by both caregivers and sick children. This might be due to frustrations of a changed life and emotional reactions [17]. By the way, this pertains as to how type 1 diabetc children must be taken care of by caregivers in different homes. For example, one may discover that some caregivers may give a sick child the wrong medication due to forgetfulness. On the other hand, a grown-up sick child may take the wrong medicine as a compromise for having done the right thing in relation to his or her caregiver. From my own perceptive, it simply leads one to an opinion that some mistakes are committed by both the caregivers and sick children, as a compromise to getting rid of the sick child's remedy. However, it could also be a reminder that those mistakes are not intentionally done but only as a compromise to try and abide by the T1DM patients' speedy recovery perceptions.

A comment done by [18] indicated that everyone could make mistakes, as there is no one who is perfect in life. Since the beginning of this section of the study, blame after blame has been aimed at elderly caregivers for some valid reasons. Regarding the T1DM rules, sick children could also be rightfully blamed. On the other hand, one finds that sick children ignore or neglect everything they might have been taught. The above quote confirms that, while at school with their friends who are not sick, they start to eat or drink forbidden foods and fruits. More importantly, it then further compels one to conclude that caregivers are full of knowledge of such instances, but the sick children are not willing to abide by the doctor's and dieticians' rules. In addition to this, some sick children could come back home being carried by their friends from schools or playing places after having eaten the forbidden foods or fruits in agreement with their peers' compulsion.

The T1DM patients only adhere firmly to the diabetic diet rules while still young but started to ignore them while at their adolescent ages. Usually, any caregiver has a chance of injecting the sick child up to the ages of 7 to 8; thereafter, such a sick child might refuse or request to be allowed to inject herself or himself. This was supported by [19], who encourages the self-management of people living with diabetes mellitus and other chronic conditions. In some instances, such a sick child could only agree to be forcefully injected after the very caregiver has called the bigger boys from the neighbors to assist her. More significantly, this is indeed not the right way of letting the caregiver work. We must always

bear in mind that the sick children are the ones who will benefit after such a treatment is done, not the caregivers. Despite that, sick children are the ones who will stay healthier or be cured after the use of whatever medication.

More significantly, the action of fearing the giving or receiving of injections is itself shaped in a dual structure that leads to either a problem or challenge. The main reason is that it is not only the sick children who fear being injected but also their caregivers; hence, they are not sure whether what they are doing is right or not. This might be an indication that the caregivers do not know the application of the injections well because they grew up without experiencing such an event [20]. In this regard, the point that they are of lower levels of education in terms of using either medications or the application of injections to their patients with T1DM disease could be the main cause. On the other hand, those sick children appear to have been fearful of the application of injections since their births. In addition to this, whenever care is conducted, be it in a hospital or clinic, if injections were given, one would always hear sick children crying at the top of their voices. In this regard, it then indeed forces one to simply conclude that any right-thinking person should never further argue why sick children always cry while they are being injected. Nonetheless, it also leads one to conclude that some sick children are readmitted to the hospitals after their caregivers have given them the wrong medications or applied injections to the wrong parts of their patients' bodies.

From the aforementioned statements, most caregivers have a poor understanding of medication and health instructions as posed by the professional health workers from hospitals. Furthermore, the reason behind this is that many of them belong to the past ages, wherein they were not forced to continue with education but compelled to engage in earlier marriages before they could reach the real ages. For instance, among those caregiver grandmothers, some of them fall under the old-age category. In light of the above-mentioned statements, the idea that some of them are illiterate is true, because they left schooling at an early age. Nonetheless, the fact that some of them are not able to even read or write their names is an undeniable truth. However, one may find all of them attending the check-up programs regularly and being taught by professional health workers seriously. The study done by [21] also reported on the challenges faced by caregivers when it came to the administration of medicine. Caregivers usually end up not being able to use the given medications properly. Having been taught that physical activities should be done at least one day a week, one would find the caregivers not even following those rules. This, in fact, means that such ignorant actions may lead to some detrimental effects on the future of the sick child as time goes on. The main reason behind that is that physical exercise, which enables a sick child to be healthy, was not done.

## 5. Conclusions

The study explored the major themes and sub-themes of the challenges faced by caregivers of children with T1DM in the Vhembe district of Limpopo province, South Africa. The main findings revealed that the lack of knowledge relating to T1DM factors on medication and diet was a major challenge facing caregivers. Similarly, caregivers expressed further challenges experienced during the provision of care for the children. The majority of children are reluctant or even ignorant in adhering to the required diet rules. Furthermore, the work commitments of the parents were identified as a concern due to the primary caregivers often releasing their roles to secondary caregivers, who were mostly grandparents who knew little about T1DM. The fear, by both children and caregivers in the administering of insulin injections, was also perceived as a concerning challenge. Therefore, the study recommended that both children and caregivers (including mother, father, teacher, grandmother, etc.) are to be taught the importance of treatment adherence, dietary management, and exercise style. Future studies related to the challenges and experiences of the children themselves should also be considered.

**Supplementary Materials:** The following supporting information can be downloaded at: https://www.mdpi.com/article/10.3390/nursrep12040085/s1, Consolidated criteria for reporting qualitative studies (COREQ): 32-item checklist.

**Author Contributions:** Conceptualization, N.S.R. and M.N.; methodology, M.N. and A.R.T.; validation, N.S.R., A.R.T. and M.N.; formal analysis, A.R.T.; investigation, M.N.; writing—original draft preparation, A.R.T.; writing—review and editing, M.N.; visualization, N.S.R.; supervision, A.R.T. All authors have read and agreed to the published version of the manuscript.

**Funding:** This research received no funding.

**Institutional Review Board Statement:** The study was approved by the Institution's Ethical Committee and the approval number was (SHS/20/PDC/32/3107).

**Informed Consent Statement:** Informed consent was obtained from all participants involved in the study.

**Data Availability Statement:** The data presented in this study are available on request from the corresponding author.

**Acknowledgments:** The authors would like to thank participants who gave consent to participate and for their cooperation during the research process.

**Conflicts of Interest:** The authors declare no conflict of interest.

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
