# Peer review of "Challenges of Caregivers Regarding Homecare to Type 1 Diabetic Children in Vhembe District, South Africa: A Qualitative Study Report"

_nursrep, doi:10.3390/nursrep12040085_

Round 1
Reviewer 1 Report (Previous Reviewer 2)
Dear authors
Thank you for the opportunity to review your article.
This manuscript has conditions to be accepted after minor revision.
Author Response
All minor comments attended to
Reviewer 2 Report (Previous Reviewer 3)
The authors have attempted changes in the manuscript since my last review, however, in my opinion the key deficiencies in this study's methodology and merit have not been addressed. The methodology is still weak. This "research" is essentially an interview of 15 people and no meaningful conclusions can be drawn.
Some other improvements needed: The conclusion in the "abstract section has been edited but the fact remains that there is nothing in this research paper that justifies the conclusion that nursing curriculum changes are needed. It is speculation.
Introduction: I am unclear why the authors mention the reference regarding sleep deprivation in a paper about diabetes.
Author Response
All the comments and clarity seeking attended to. Thanks

Reviewer 3 Report (New Reviewer)
The paper is crucial for the management of children affected by diabetes. The authors can discute the educational role of different caregivers (including mother, father, teacher, grandmother etc) for the dietary management and the exercise style.
However, the authors can describe the educational role of caregiver for management of acute complications of diabetes.
Author Response
Comments have been attended to
This manuscript is a resubmission of an earlier submission. The following is a list of the peer review reports and author responses from that submission.
Round 1
Reviewer 1 Report
Ndou M et al. exploring the challenges experienced by caregivers during the provision of care to Type-1 diabetic children. The study is certainly interesting but the main problem with this study is the assessment of Caregivers experience. In literature, caregivers experience has been analysed using Informal Caregiver Burden Assessment Questionnaire (QASCI). The QASCI has already been validated in caregivers of people with various chronic diseases, presenting good validity, test-retest reliability and internal consistency results. It has 32 items scored from 1 to 5 and consists of the following 7 subscales: Emotional Burden (4 items), Implications for Personal Life (11 items), Financial Overload (2 items), Reactions to Demands (5 items), Perception of Efficacy and Control Mechanisms (3 items), Family Support (2 items) and Satisfaction with the Role (5 items). I suggest using this questionnaire and resubmit the article
Reviewer 2 Report
Dears authors
Thank you for the opportunity to review your article.
Brief summary: This study exploring the challenges experienced by caregivers during the provision of care for Type-1 diabetic children and the requirement nursing curriculum should incorporate home visits to assist caregivers during the provision of home care for Type 1-Diabetic children.
The findings have important implications not only for research but also for clinical practice. It allows the development of critical thinking in nursing and, in addition, contributes to the interventions in this area.
Areas of strength
The references included are relevant for the subject under study although it only presents 3/16 (19%) references are not from the last 5 years. There is strong concordance between the objective and the methods used. The description of the methodology was made in a clear and adequate way. The results are clearly described. The discussion correlates with the presented data and takes the published literature into account. The manuscript presents some limitations and clinical implications.
This manuscript has conditions to be accepted in its current format.
Congratulations to the authors for their work
Reviewer 3 Report
Raliphaswa et al present a study on a very important topic, but there are several deficiencies in the study.
1. Title and abstract need to mention what country the study was conducted in.
2. Quality of English writing needs significant improvement.
3. Data is very limited and non-representative: All the caregivers in the study sample are single or widowed, with complete absence of married individuals. It can be argued that single caregiver households would have more challenges regardless.
4. Most participants are on governmental subsidy, again concerning that the sample is non-represntative and predominantly lower socioeconomic class.
5. Discussion comes across as age-ist and discriminatory against elderly people.
6. The conclusion in abstract states that "nursing curriculum should incorporate home visits". However, nothing in the manuscript supports this conclusion.
Reviewer 4 Report
Thank you for allowing me to review this manuscript. This manuscript entitled "Challenges of caregivers in home care of children with type 1 diabetes in Vhembe district" aim to explore the challenges experienced by caregivers during the care of children with type 1 diabetes.
It is an interesting and highly relevant article today, although it has several limitations that make it suitable for publication in this journal. These limitations are detailed below:
- In the introduction, it would be important to point out the relevance and timeliness of the subject of study. Also, enough citations are missing. I would recommend expanding this section by citing more expert authors on the subject and expanding this section of such relevance. Also, it would be important to clearly state the objective of the study.
- In the material and methods section it is described in detail. However, as a signal and weakness, we can highlight that different important considerations are not reflected. It would be necessary to indicate whether the total sample is representative and allows the generalization of results. In addition, it would be important to highlight the inclusion and exclusion criteria of this research.
- The results are presented in a clear and orderly manner. Also, an interpretation of them is reflected. However, in the tables the acronyms used in them are not specified at the foot of the table.
- In relation to the discussion, in the manuscript the results are discussed in an orderly manner. However, due to the importance of the subject of study, the number of studies cited in this section will be modified, which will allow justifying the results and comparing them. We recommend expanding the number of citations that would allow discussion of the results presented, providing more scientific rigor to the article presented. Also, it would be important to point out limitations that have been found during its preparation.
- In relation to the conclusions, they are clear and precise. However, we recommend indicating more precisely the implication in clinical practice of the same. Also, it would be interesting to state the lines of the future that the authors consider.
Good work